# Research on the Thermal Properties of Fireplace Concrete Materials Containing Various Mineral Aggregates Enriched by Organic and Inorganic Fibers

**DOI:** 10.3390/ma14040904

**Published:** 2021-02-14

**Authors:** Agata Stempkowska, Joanna Mastalska-Popławska, Piotr Izak, Łukasz Wójcik, Tomasz Gawenda, Marzena Karbowy

**Affiliations:** 1Faculty of Mining and Geoengineering, AGH University of Science and Technology, Mickiewicza 30 Av., 30-094 Krakow, Poland; gawenda@agh.edu.pl; 2Faculty of Materials Science and Ceramics, AGH University of Science and Technology, Mickiewicza 30 Av., 30-094 Krakow, Poland; jmast@agh.edu.pl (J.M.-P.); izak@agh.edu.pl (P.I.); lukwoj@agh.edu.pl (Ł.W.); 3Northstar Poland, 27 Stycznia 47-48 St., 64-980 Trzcianka, Poland; marzena.karbowy@northstar.pl

**Keywords:** concrete, thermal properties, structural properties

## Abstract

This work presents a summary of research on concrete fireplace materials made of various mineral aggregates and enriched with steel and organic fibers. To determine the optimal applications of such concretes, their ability to accumulate heat and their other physicochemical parameters were tested and analyzed. Studies on the behavior of concrete materials during cooling are reported, and the ability of such materials to accumulate heat is evaluated using calculations. In addition, tests were performed on the loss of mass during heating, as well as on the mechanical bending strength and microstructures of these materials. Studies have shown that the behavior of concrete materials at high temperatures can be modified and adapted for specific purposes. The addition of fibers to concrete improves both the mechanical properties of mortars and the heat flow in concrete materials.

## 1. Introduction

Over recent decades, the energy demands in residential buildings have increased, and there is now a great need to balance energy consumption. Energy-efficient buildings are becoming increasingly desirable due to rising energy costs and increasing awareness of global warming [1]. The thermal properties of concrete materials are attracting increasingly more attention, not only because of their influence on the energy efficiency of buildings but also due to the structural properties and functionality of such materials. Modern concrete materials containing various complementary cement materials, various types of aggregates (including light and recycled aggregates), and fibers are increasingly used in transport structures such as sidewalks and bridge decks, as well as in large foundations (ready-mixed concrete), where thermal behavior is important and sensitive to construction properties [2]. However, there are few publications on the heat-accumulation properties of concrete elements used in fireplaces. Usually, the attention is focused on the role of the thermal conductivity coefficient in the insulation of buildings, and information can also be found on the behavior of concrete at high temperatures above 500 °C, for which the strong influence of the aggregates used and effective reinforcement with steel and polypropylene fibers are highlighted [3]. The type of aggregate is also important in concretes operating at increased temperatures. Fine aggregates based on sand show a polymorphic transformation at 573 °C and have a high coefficient of thermal expansion, which causes micro-cracking in the material and weakening of the structure [4]. Interesting results were also obtained by examining the compressive strength and pore distribution in concrete working at above 500 °C [5,6]. The addition of expanded glass to concrete also significantly reduces thermal conductivity [7], so such materials have good heat accumulation capacities [8]. This publication focuses on concrete materials designated for heating devices. Heating furnaces are usually made of natural soapstone (a green–grey rock containing mainly magnesium aluminosilicates), which has a very good thermal accumulation capacity. In these furnaces, thanks to the accumulation capacity of soapstone and technical solutions (e.g., multiple circulation and swirling of flue gases), the temperature in the furnace reaches about 1200 °C (normally about 600 °C), which enables the combustion of flue gases (burning of soot). This results in high furnace efficiency and has a positive impact on the environment [9]. The temperatures in accumulation furnaces reach 400 °C and higher. Simple concrete is rarely used as a material in these furnaces due to its low resistance to such temperatures (the safe temperature limit is 450 °C). This is why ceramics are used in these applications, as ceramics are much more resistant to high temperatures, especially chamotte. The behavior of cement mortars at high temperatures can also be modified and adapted for specific purposes. For this purpose, it is worthwhile to trace the physicochemical processes that occur under the influence of high temperatures.

To control the process of thermal accumulation, it is necessary to determine the thermal phenomena and transformations taking place in the materials. The energy absorbed by a body during heating or lost during cooling is proportional to the product of the body weight *m* and the temperature difference of the body Δ*T* before and after the thermal transformation. The principle of thermal conversion ability Δ*Q* can be rendered as [10]:*ΔQ* = *c*_v_ × *m* × Δ*T* (J)(1)
where the *c_v_* capacity of a substance is considered the amount of heat needed to raise the temperature of the substance by one degree. The thermal capacity, which is measured per unit of mass of a substance, is called specific heat *c*_p_ (expressed in J/kgK). This quantity is not a constant value and depends primarily on temperature (Figure 1) [11]. In many amorphous, glassy, and crystalline substances, the specific heat increases simultaneously with a temperature increase, as well as at high temperatures.

Another value that characterizes materials in terms of their thermal properties is their thermal volume capacity. The value *b* is calculated as the product of the thermal capacity *c*_v_ and the density *ρ* of the material from which the body is made:*b = c*_v_ × *ρ* (J/(m^3^K))(2)

The values of thermal capacity *b* for building materials can be very different, ranging from about 25 kJ/m^3^K for insulation materials to almost 3000 kJ/m^3^K for heat accumulators. For example, for ordinary chamotte bricks, the thermal energy accumulated in a unit of volume is about 1000 kJ/m^3^K, while for concretes resistant to high temperatures, with the same volume, much more energy can be accumulated (~2700 kJ/m^3^K). In general, heat-resistant concretes can accumulate about three times more energy than ordinary chamotte coverings. The main factor influencing the value of concrete is its composition, as this capacity for concrete results from the volumetric thermal capacity of its ingredients, which are additive quantities [10,11].

Volume heat capacity is the amount of energy taken in during heating or released during cooling 1 m^3^ of the material, changing the material’s temperature by one degree. In other words, volume heat capacity is the energy that increases (or decreases) the temperature of a given material of a unit volume by a unit of temperature. Volume heat capacity is not sufficient, however, to describe the ability to accumulate heat. An additional parameter characterizing the efficiency of the material accumulation phenomenon is the energy that can be accumulated in a unit of the material’s volume for a certain period of time Δ*t*. The maximum energy that can be accumulated in a given *b*_max_ material can be described by the following formula [11]:*b*_max_ = *b* × Δ*t* (J/m^3^)(3)

It follows that different building and mineral materials have different capacities to store energy, the value of which determines if the material is a good heat accumulator and can be used as such in space-heating systems. An important element for thermal properties is the use of natural [3], light [12,13], hybrid [14], or recycled [15,16] aggregates. The properties of concrete materials with rubber waste particles were also studied [17]. In the literature review, studies on the thermal properties of concretes and mortars containing recycled glass as fine aggregates [18] and reinforced concretes containing steel, plastic, and glass fibers [19] were found. Concrete is a multi-phase and complex system. Therefore, numerical methods based on multi-species modeling seem very promising for the analysis of concrete [20]. Numerical simulations have been widely used to analyze the influence of aggregates on the transport properties in concrete, including heat transport. These methods allow one to predict the important properties of concretes such as their resistance to chloride corrosion [21], which is significantly influenced by temperature [22], as well as the shapes of the aggregate grains [23,24].

Another parameter that determines the thermal properties of materials is the time needed to release (emission) the accumulated energy. With a given amount of accumulated energy, the emission time of that energy must not be too short (or too much heat will be released in a unit of time) or too long (or too little heat will be released in a unit of time and will be insufficient, e.g., for heating the room). In a thermodynamic system (e.g., in an isolated room) containing a body with a temperature higher than room temperature, entropy will strive to achieve the maximum value, i.e., the equilibrium state of the whole system, to equalize the temperature in the whole area. This process always occurs automatically and spontaneously. The phenomenon of heat transfer from the body to the system is called the emission of heat energy. The measure of thermal energy emission *E* is the thermal power *P*, which is determined by the following relation [25]:*P* = *E/t* (J/s).(4)

Heat-accumulating materials used in practical applications should have a relatively high heat power. The heat that is delivered to the storage material then causes a proportional increase in temperature. Sensible heat is presently the most popular method of thermal energy storage. This change can be registered with the senses or via sensors [26]. Sensible heat magazines accumulate thermal energy by heating or cooling the stored material. These magazines then use the thermal capacity and temperature changes of the material during the charging and discharging processes. The amount of accumulated heat depends on the mass and medium of heat used for storage and the temperature difference between the initial and final states [27,28]. There have also been attempts to model the behavior of concrete materials under the influence of high temperature and to estimate their mechanical properties and heat accumulation capacities [29,30].

## 2. Research Significance

Materials used for the construction of fireplaces should have special thermal properties which can allow for the accumulation and release of heat. Heat accumulation refers to the ability of a material to accumulate and store a certain amount of thermal energy inside itself, which can then be released for a period of time by the material. That is why we are looking for ways to accumulate heat when there is excess and use it when there is a deficiency. Foundationally, this involves finding optimal solutions from the perspective of so-called thermal comfort, i.e., obtaining and maintaining the necessary temperatures inside objects for comfortable functioning in unfavorable external conditions. One example is seasonal accumulation, which involves storing heat energy in summer and using that energy in the autumn–winter season. Reverse processes are used to cool rooms in summer conditions. The purpose of this work is to try to determine whether concretes made with mineral aggregates and various organic additives have different heat accumulation capacities. This would allow the concrete to be used not only for the manufacturing of elements, including prefabricated elements, at increased temperatures but also where such elements could accumulate and release heat (e.g., fireplace covers). Therefore, the basic parameters characterizing the usefulness of such materials, i.e., volume and mass thermal capacity, maximum heat accumulation capacity, and high temperature behavior (differential thermal analysis(DTA) tests), were assessed.

## 3. Materials and Methods

### 3.1. Materials and Mix Proportions

Samples for testing were supplied from Northstar and were used as fireplace insulation inserts. The composition of the individual concrete materials is shown in Table 1.

### 3.2. Research Methodology

#### 3.2.1. Thermal Properties

There are many methods for testing the thermal properties of concrete materials [31,32,33]. In this publication, studies were carried out with the use of a thermal imaging camera NEC ThermoGear G100, which enabled the recording and visualization of temperature distribution on object surfaces (mapping thermal images of the objects). This system works on the principle of processing the infrared radiation emitted or reflected by these objects into an electrical signal and then into the image viewed on the screen as a so-called thermogram. Thermal imaging enables the detection of many properties of plastics in a way that no other technology provides. Samples of 10 cm × 10 cm were cut out of the tested concrete. Then, the samples were placed in a laboratory dryer and heated to 160 °C for 2 h. Next, the samples were taken out, immediately placed on a pedestal (Figure 2), and the temperatures were measured based on the cooling time. These tests were carried out with the use of a thermal imaging camera that was placed 70 cm from the tested sample.

#### 3.2.2. Differential Thermal Analysis

Concrete plastics and gas emissions were examined using an STA 449 F3 Jupiter Thermal Analyzer (Netzsch) and a coupled quadrupole mass spectrometer TA-QMS Coupling (Netzsch). The measurements were carried out in alumina crucibles at a heating rate of 10 °C/min in a temperature range of 30–600 °C under an air and argon atmosphere with a constant flow of 20 mL/min. This research focused on the analysis of concrete materials 2, 3, and 5 and the influence of additives modifying the strength properties in terms of gas emissions. 

#### 3.2.3. Mechanical Pproperties

Samples for mechanical strength testing were prepared in accordance with PN-EN 206 + A1:2016–12 [34], and bending strength measurement was performed after 1, 7, and 28 days of maturation. These measurements used bars with rectangular cross-sections and dimensions of 40 × 40 × 160 mm^3^. The mechanical strength of the tested fireplace materials was determined using the three-point bending method. The specimen was placed in a special holder in a testing machine (Zwick/Roell 2.5, Ulm (Germany)) (Figure 3). A measurement was then performed three times.

#### 3.2.4. Microstructure of the Fireplace Concrete

Samples of the delivered concrete materials were taken at random and analyzed on a scanning microscope equipped with an energy dispersive spectroscopy(EDS) analyzer (Ultra High Resolution Scanning Electron Microscope with Field Emission Gun (FEG)–NOVA NANO SEM 200 (manufacturer FEI EUROPE COMPANY), cooperating with an EDS analyzer by EDAX). Photographs at a magnification of about 100,000 (20,000) times are shown in Figures 5–10. Under each photograph, the EDS analysis is shown on the marked points.

## 4. Results

### 4.1. Thermal Properties

To standardize the data needed to calculate, among others, thermal parameters, the mass, volume, and density of the samples were determined (Table 2).

Table 3 outlines the temperatures obtained depending on the cooling time. At the beginning of the measurement, the temperature between measurements was measured every few seconds, every 30 s, and then every 1 min. Using the thermal imaging camera, it was possible to implement a program for reading the temperature values on the surfaces of the samples in different places (Figure 4). To calculate the thermal capacity, the value a is used, i.e., the temperature measured in the middle of the sample.

Figure 5 shows a diagram of temperature changes (value a) on the surfaces of the cooling samples.

#### Calculation of Heat Capacity

The specific heat is an additive quantity, i.e., each degree of freedom in a given system contributes to the total specific heat of the system, and, as a result, the total specific heat is the sum of the different contributions [35]. On the basis of the percentage content of individual raw materials (Table 1) and their table values of *c*_v_ specific heat (Table 4) [36], the total specific heat of the examined materials was calculated:*c*_v_ = *Σ*%*x* × *c*_vx_(5)
where %*x* is the percentage of raw material in the sample [-], and *c*_vx_ is the table-specific heat value (J/kgK).

The calculations are presented in Table 5.

On the basis of the value of total specific heat *Σ c*_v_ (Table 5), the mass of the tested samples *m* (Table 2) and ∆*T* (the difference in the temperature of a given body before and after thermal transformation; Table 6) and ∆*Q*, i.e., the change in the thermal energy of the tested materials, were calculated. The results of the calculations are presented in Table 6. 

Another value that characterizes the thermal properties of materials is volumetric thermal capacity b, which describes the ability of amaterial to accumulate heat. The calculated values of the volumetric thermal capacity are shown in Table 7.

When describing the thermal properties of fireplace materials, it is also important to know the thermal power of the tested materials *P*, i.e., the parameter determining the transfer of heat from the body to the system. The relevant calculations are presented in Table 8.

In the initial cooling phase (up to 10 min), all analyzed samples lost heat at a similar rate; then, the trend changed. The curves in Figure 5 and the temperature values in Table 3 indicate that sample no. 4 (with 60.477% weight of sand content) lost heat the fastest, while sample no. 5 (with 32.027% weight of sodium–calcium feldspar content) lost heat the slowest.

Beyond the different compositions of the samples, this phenomenon could also be influenced by the fact that the samples differed significantly in their density. The density of sample no. 4 was 1.50 g/cm^3^, while that of sample no. 5 was 2.16 g/cm^3^. The values of their thermal parameters (thermal power, thermal volume capacity, and thermal energy), however, were found to be similar. The highest values were recorded for sample no. 5 consisting of magnesium silicates and sodium–calcium feldspar. The weakest thermal parameters were found for sample no. 4, as well as sample no. 2, which consisted mainly of cement, sand, and refractory aggregates. This is consistent with the literature data. For example, materials with the highest specific density always have the highest thermal capacity [37]. The thermal capacity of metals with a density of 7000–9000 kg/m^3^ is 1,500,000–3,500,000 J/(m^3^K). Casting rocks used as aggregates in concretes have an even smaller thermal capacity than metals (granite is about 1,800,000 J/(m^3^K)). Brick and sand have an even lower volumetric heat capacity at about 1,200,000 J/(m^3^K). For normal and refractory concrete with a density of about 2400 kg/m^3^, the volume heat capacity is about 2,770,000 J/(m^3^K) [38,39].

### 4.2. Differential Thermal Analysis

The thermal curve differential thermal analysis/thermal gravimetric analysis (DTA/TG), differential scanning calorimetry (DSC) analysis, and total ion current (TIC) gas emission analysis are presented in Figure 6, Figure 7 and Figure 8. Materials containing organic fibers were selected for these analyses (Table 1). Based on the performed tests, it can be concluded that the emission of the detected gases is related to the gases supplied to the sample during the sample’s heating, i.e., O_2_(32) and N_2_(28), as well as Ar(39), introduced as an inert gas to protect the microbalance. The emission of water was observed in all of the samples, and the amount of water changed successively during the heating of the samples. The highest amount of water was observed at around 140 °C, and the lowest amount, related to the dehydroxylation of the cement products, was observed at 480 °C. In the tested samples, weight loss in the temperature range of 30–600 °C was also observed. The complex endothermic effect observed in the temperature range from about 50 to about 350 °C is related to the dehydration of silicates of type C–S–H, hydrated calcium aluminates and aluminosulphates, and the decomposition of gypsum. Another endothermic effect in the temperature range from about 400 to about 410 °C is attributed to C_2_SH decomposition (2CaO·SiO_2_·H_2_O). The endothermic effect in the temperature range from about 490 to about 510 °C is attributed to Ca(OH)_2_dehydroxylation. The total weight loss of the samples varied and was 10.2% for sample 2, almost 7% for sample 3, and almost 9% for sample 5. This is due to differences in the composition of the individual materials.

The observed mass loss is different in particular types of concretes. These differences most likely result from different aging times, i.e., the time from concrete preparation to measurement (different humidity values), and different amounts and types of aggregates. In general, it can be concluded that the addition of modifiers causes an increase in the temperature of the cycle in which the DTA/DSC analysis takes place and causes an increase in the weight loss. Organic additives likely facilitate the emission of water at about 140 °C.

The tests showed that the specific heat of the concrete material strongly depends on the amount of free water that is released in the temperature range of 50–90 °C. Values of the specific heat measured by this method significantly drop at higher temperatures, which is likely related to the absorption of heat during the dehydration reactions of concrete materials.

### 4.3. Mechanical Properties

Table 9 shows the maximum values of the mechanical strength under bending for the tested fireplace materials after the different aging periods.

Samples 1 and 3 showed the highest mechanical strength at more than twice that of the other tested materials. Their bending strength after 28 days of aging was 12.3 and 13.2 MPa, respectively. The main additives in both samples were cement and aggregate. As a reinforcing additive, sample 1 included glass fibers, while sample 3 used glass and polypropylene fibers. Samples 2, 4, and 5 included a minor addition of steel fibers, which are also used as a strengthening additive to maintain the concrete’s structure during operations at high temperatures, causing significant stress in the system. Sample 5, despite the addition of reinforcing fibers, was characterized by low mechanical strength (4.8 MPa) due to the addition of a mafic aggregate containing more than 2% free biotite. Increased mica content causes a significant and constant decrease in compressive strength [40], which is especially important in the case of the fine aggregates (e.g., mafic sand) [41].

### 4.4. Microstructure of Fireplace Concrete

Scanning electron microscope(SEM) microphotographs with the energy dispersive spectroscopy (EDS)system of the tested samples are shown in Figure 9, Figure 10, Figure 11, Figure 12 and Figure 13. Based on the conducted research, it is possible to determine the compact microstructure of all of the concrete materials, except for sample 4, which is characterized by the presence of large spherical pores (Figure 12). A compact structure was also present in the concrete materials containing glass fibers (Figure 9) or polypropylene fibers (Figure 11, Figure 12 and Figure 13). Generally, a homogeneous microstructure was found in the analyzed concrete materials; however, in the case of sample 5, the uneven distribution of fibers was noted (Figure 13). There were also visible pores (except for sample 4), which may have resulted from inappropriate selection of the aggregate graining (filler) of the concrete material or deaeration. This is an important factor because the presence of pores lowers the density and thermal properties of the concrete material. EDS measurements confirmed the presence of cement-derived aluminates and aluminosilicates, as well as the presence of fibers, aggregate grains, and pigments.

Quartz grains and zirconium were present in the microstructure of sample 1 (Figure 9). Sample 3 (Figure 11) contained increased amounts of calcium aluminosilicate and strengthening organic fibers. Sample 4 (Figure 12) is a foamed concrete featuring the addition of calcium carbonate. In the composition of sample 2 (Figure 10), increased amounts of magnesium, zirconium, and calcium can be seen. In particular, in micro-area b, dolomite crystals formed with the addition of titanium white reinforced with organic fibers were found. Concrete no. 5 (Figure 13) is a typical concrete reinforced with organic fibers and aggregate, where the presence of iron comes from the minerals of the aggregate.

From an application perspective, concrete systems can be reduced to a mixture of three components: cement, fine aggregate, and water. Cement is a composite system in which aggregate grains are surrounded by a hardened cement slurry. Concrete formation is determined by the cement’s hydration reactions. Cement is a mixture of solid phases and mainly contains various types of aluminates and calcium aluminosilicates: allite C_3_S-(3CaO·SiO_2_), bellite C_2_S-(2CaO·SiO_2_), tricalcium aluminateC_3_A-(3CaO·Al_2_O_3_), brownmilleriteC_4_AF-(4CaO·Al_2_O_3_·Fe_2_O_3_), calcium oxide (free)-CaO, and gypsum-(CaSO_4_·0.5H_2_O). 

Cement hydration is a multi-stage process. However, it can be assumed that the main phases of hydration are as follows:allite:  3CaO·SiO_2_ + 3H_2_O → C_1.5_SH_1.5_ + 1.5·H_2_Obellite:   2CaO·SiO_2_ + 2H_2_O → C_1.5_SH_1.5_ + 0.5 H_2_Ogypsum:  CaSO_4_·0.5H_2_O + 1.5H_2_O → CaSO_4_·2H_2_Ocalcium oxide:  CaO + H_2_O → Ca(OH)_2_(6)

The hydration of tricalcium aluminate, however, is more complicated. There are three basic stages in this process. In stage I, under the initially high concentration of gypsum, ettringiteis formed according to the following reaction:3CaO·Al_2_O_3_ + 3CaSO_4_·2H_2_O + 26H_2_O →3CaO·Al_2_O_3_·3CaSO_4_·32H_2_O.(7)

In stage II, when gypsum is lacking, a transitional phase is formed:3CaO·Al_2_O_3_·Ca(OH)_2_·12H_2_O.(8)

In stage III, the second stage phase reacts with ettringite, giving the low-sulfate form of aluminum sulphate according to the following reaction:3CaO·Al_2_O_3_·3CaSO_4_·32H_2_O + 3CaO·Al_2_O_3_·Ca(OH)_2_·12H_2_O → 3CaO·Al_2_O_3_·CaSO_4_·12H_2_O + 2Ca(OH)_2_ + 20H_2_O.(9)

The hydration of braunmillerite is similar to that of tricalcium aluminate. In the presence of gypsum and water, the following structures are formed:4CaO·Al_2_O_3_·Fe_2_O_3_ + CaSO_4_·2H_2_O + Ca(OH)_2_ + aq → 3CaO·(Al_2_O_3_, Fe_2_O_3_)·3CaSO_4_·32H_2_O.(10)

After braunmillerite is exhausted, the structure becomes 3CaO·(Al_2_O_3_,Fe_2_O_3_)·CaSO_4_·12H_2_O [42,43].

The hardened cement slurry is largely dominated by two phases: the CSH gel phase and the portlandite (Ca(OH)_2_) phase. There are also additional phases resulting from the hydration of tricalcium aluminate and braunmillerite. It should be noted that these phases contain large amounts of crystalline water, and this water is the main destructive factor for cement mortars at high temperatures, which accompanies the use of fireplaces and heating stoves. However, the content of elements such as magnesium, aluminum, calcium, and silicon significantly affects the thermal properties of the concrete [44].

## 5. Discussion

The addition of fibers to concrete improves the mechanical properties of mortars and may improve heat flow in concrete materials. At elevated temperatures, polypropylene melts and is absorbed by the surrounding cement matrix, creating a network of channels through which moisture and the resulting water vapor can flow [45]. Hardened cement mortar can also be treated as a fibrous composite in which the matrix is the hardened cement mortar. In the case of such a composite, the addition of a small amount of fibers, even with an elasticity modulus lower than that of the matrix, significantly improves the mechanical properties of the mortar. This is because during the hydration process of the paste components, a number of reactionsoccur, leading to the formation of mechanical stresses. The resulting stresses, which are related to internal contractions, are leveled until the structure stiffens. However, in the initial stage of hardening, the hardening paste has too low a mechanical strength to compensate for the stresses accompanying the hydration process (as confirmed by the results presented in Table 9). 

The results of the mechanical strength after 28 days confirm this explanation. The highest bending strength was found for the samples with the highest amount of glass and polypropylene fibers, i.e., sample 1 (0.266 wt % of glass fibers) and sample 3 (0.209 wt % of glass fibers and 0.02 wt % of polypropylene fibers). The addition of fewer of these fibers or the addition of steel fibers (even in combination with glass fibers) caused more than a twofold decrease in mechanical strength in the described case, from 13.2 MPa for sample 3 to 4.7 MPa for sample 4.

When the proportion of organic fibers is so large that the fibers form a self-connected network of channels, it is known as a parallel transport model. That is, the transport of liquids and gases at temperatures above 400 °C takes place both through the network formed after melting the fibers and through the pores and cracks existing in the material. When there are fewer fibers, transport can be described by the so-called series-parallel model, where the incompletely connected network of channels is supplemented by the existing defects in the material [46,47].

The introduction of short polypropylene fibers, in addition to preventing detonation splashes during rapid heating or contact with fire, also has a very positive effect on the mechanical properties of mortars. Hardened cement mortar can be treated as a fibrous composite in which the matrix is hardened cement paste. In the case of such a composite, the addition of a small amount of fibers, even with a modulus of elasticity lower than that of the matrix, results in a significant improvement in the composite’s mechanical properties. This is because several stress reactions take place during the hydration of the cement slurry components. In the subsequent period, the stresses arising from internal contraction are leveled until the structure is stiffened. However, in the initial stage of hardening, the hardening cement slurry still has too low a mechanical strength to compensate for the stresses accompanying the hydration. This yields the formation of a micro-fracture network, leading to material defects and a loss of continuity at the microscale. The addition of fibers causes bridging of the resulting cracks, which reduces their size and improves the material’s resistance to brittle fractures [48].

However, the addition of short fibers to the cement matrix may regulate more than the matrix’s hydration and hardening properties. Through the addition of fibers with an elasticity modulus higher than the matrix modulus, the strength properties of the material can be directly modulated. The addition of such fibers ensures that the material does not disintegrate when the cement matrix breaks, allowing the matrix to still transfer some loads.

The reactions taking place in the cement during heating can be divided into five stages:I.At a temperature of about 100 °C, mechanically (capillary) bound moisture evaporates.II.At a temperature of about 200 °C, physicochemically bound water is released due to adsorption.III.At 400 °C, the release of chemically bound water in hydrated aluminosilicate compounds begins. This does not yet entail a significant decrease in mechanical strength.IV.At a temperature of about 530 °C, due to the decomposition of calcium hydroxide, mechanical strength decreases rapidly.V.At a temperature of about 1000 °C, the water is completely removed, and the structure disintegrates into a powder. The material sinters at even higher temperatures.

The effect of high temperature, i.e., that above 100 °C, on hardened cement material is very specific and causes the rapid desorption of moisture from the outer layers. The resulting water vapor flows towards the colder inner layers, where it is reabsorbed. As the temperature increases, the thickness of the heated layer increases. When the moisture-saturated layer does not move quickly enough, it is overtaken by the wandering temperature front, which leads to the evaporation of water at the front border and an increase in internal pressure. This creates tensile forces perpendicular to the temperature front, which in turn leads to rapid removal of the surface layers in the form of microcracks and spatters. This phenomenon mayoccur cyclically, destroying increasingly deeper layers of the material.

To counteract this phenomenon, it is necessary to improve the heat flow, which will entail a reduction in the pressure inside the heated element. The best way to do this is to create an interconnected network of channels in the material through which moisture and the resulting water vapor can flow, or toimprove the material’s thermal conductivity.

This can be achieved via the introduction of a fibrous material that melts at an elevated temperature and is easily absorbed by the surrounding cement matrix. Polypropylene is perfect for this purpose. This material melts at 170 °C, which is below the temperature at which water vapor is rapidly released and does not damage the concrete matrix. At low temperatures (<100 °C), all forms of water in concrete (capillary, adsorptive, and crystalline) become very good heat accumulators (the heat of water evaporation is approximately 80 kcal/mol). Exterior cladding concrete shapes used in furnaces are a good choice for this purpose.

Another type of thermal accumulation involves using the excess heat energy generated inside the fireplace or the stove to heat rooms over a longer timeframe. Such furnaces are made of natural soapstone (a greenish-gray rock containing mainly magnesium aluminosilicates), which has a very good heat accumulation capacity. In these furnaces, due to the accumulation capacity of soapstone and various technical solutions (e.g., the multiple circulation and swirling of exhaust gases), the temperature in the furnace can reach about 1200 °C (normally about 600 °C), which allows the so-called afterburning of exhaust gases (soot combustion). This results in high efficiency of the furnace and, above all, has a positive impact on the environment.

The temperatures in storage furnaces can reach 400 °C and higher. Ordinary concrete is rarely used as a material for furnaces due to its low resistance to such temperatures (the safe limit temperature is 450 °C). This is why, as a rule, ceramics that are much more resistant to high temperatures are used for this application, most often fireclay. In our case, sample 5 showed the highest thermal power (Table 8) (19.70 W), whereas sample 2 showed the lowestthermal power (8.55 W). In addition to the significant effect of density (sample 5—2.16 g/cm^3^; sample 2—1.76 g/m^3^), composition itself plays an important role. Sample 2 consisted mainly of cement, sand, and refractory aggregate, whereas sample 5 consisted mainly of magnesium silicates and sodium–calcium feldspar. Therefore, it is worth analyzing, in each case, the physicochemical processes thatoccur during the formation of hydration materials and their destruction under the influence of high temperatures.

## 6. Conclusions

Concrete elements can be successfully used in the construction of domestic fireplaces, thus taking advantage of the good heat accumulation properties of these elements. Both thermal and strength properties can be modified significantly via the addition of various aggregates as well as propylene and steel fibers. This modification consists mainly in changing the mechanisms of the heat flow and the decomposition products of concrete elements during heating—mainly water, although the density of the materials also has a significant impact.

From the perspective of thermal energy use, our study showed the clear advantage of concrete materials containing magnesium and sodium–calcium aggregates (feldspars). The highest values were recorded for sample 5 (which contained over 70% of the listed components). In turn, from the perspective of mechanical properties, samples 1 and 3 showed the highest mechanical strength and were reinforced with glass and polypropylene fibers, respectively. The results, however, depend on the applied temperature range. Therefore, in the case of fireplaces, it is suggested to use different multi-layer systems of concrete materials to enable the long-term heating of rooms via heat accumulation.

## Figures and Tables

**Figure 1 materials-14-00904-f001:**
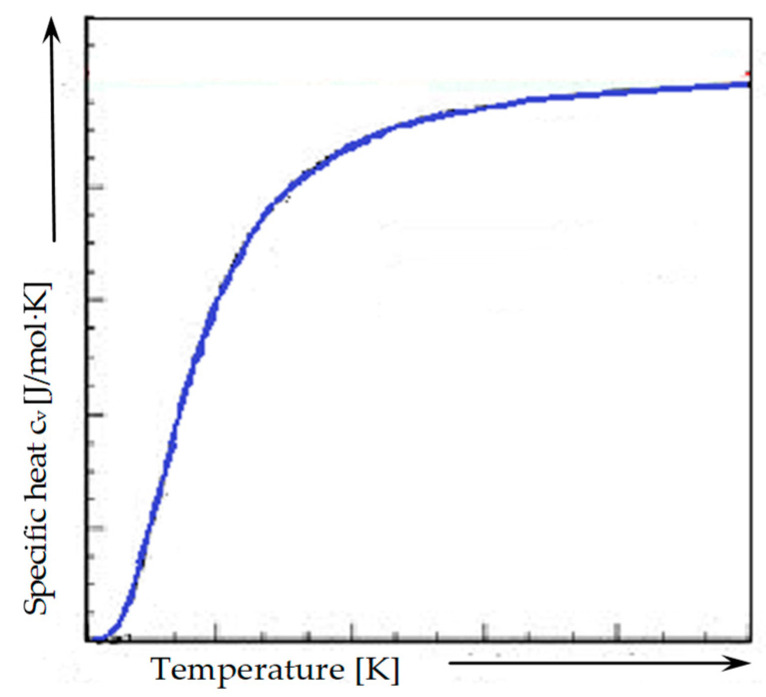
Thermal capacity dependence in function of temperature.

**Figure 2 materials-14-00904-f002:**
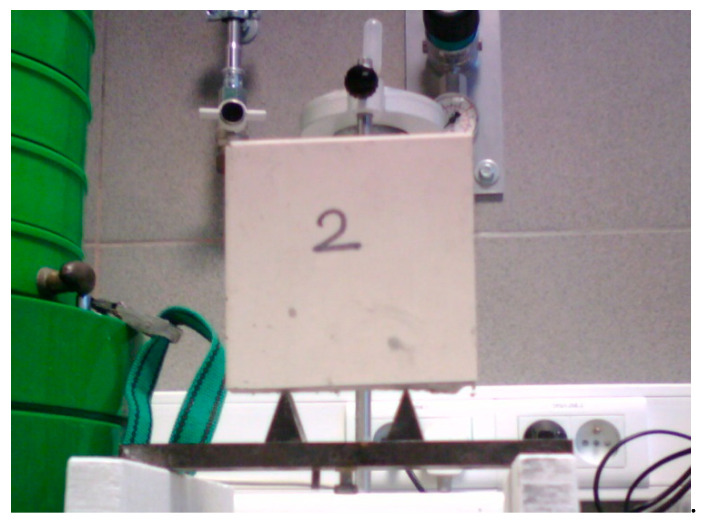
Sample during the thermal imaging test.

**Figure 3 materials-14-00904-f003:**
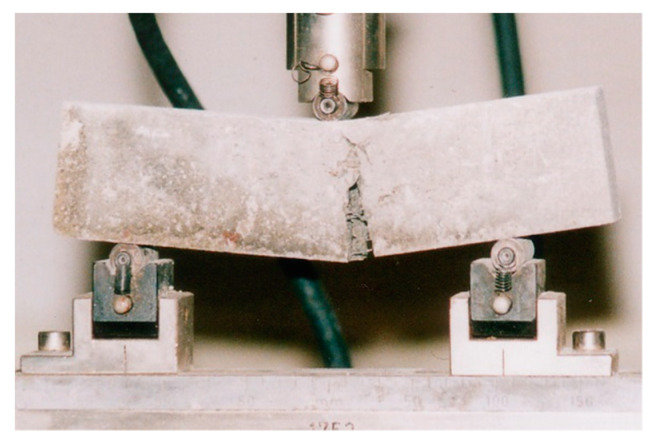
Measurement of the bending strength via three-point bending.

**Figure 4 materials-14-00904-f004:**
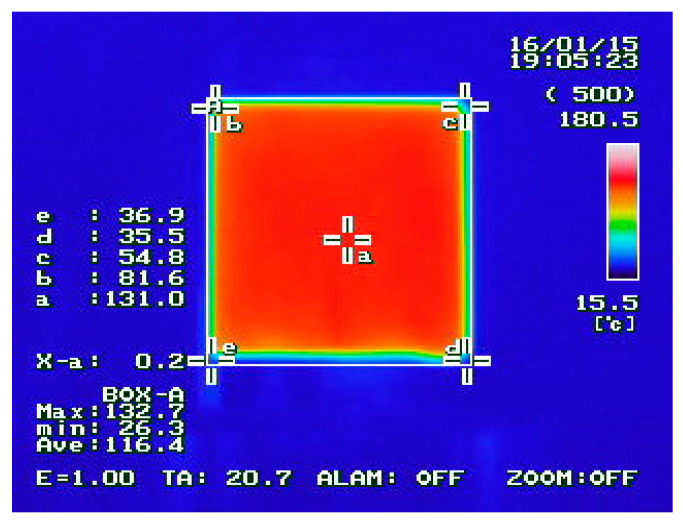
Example thermal image of the temperature distribution on the surface of the tested sample.

**Figure 5 materials-14-00904-f005:**
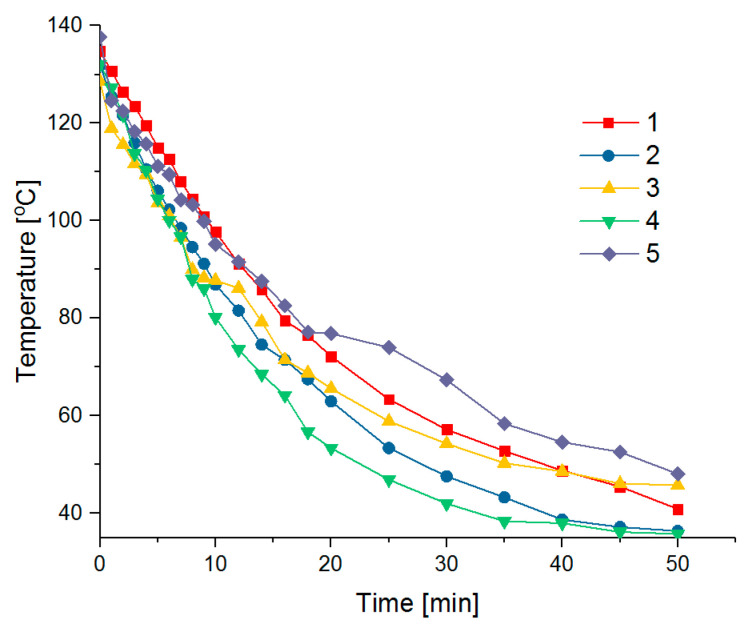
Combined temperature distribution diagram on the surface of cooling samples.

**Figure 6 materials-14-00904-f006:**
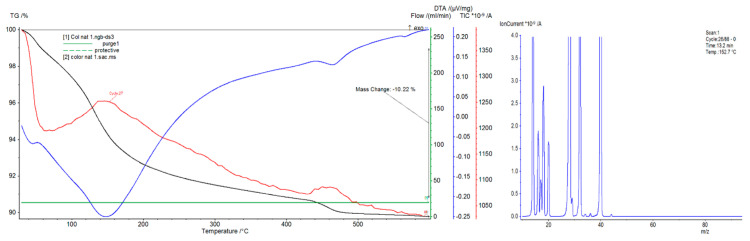
Thermal analysis differential thermal analysis/differential scanning calorimetry (DTA/DSC) of sample 2.

**Figure 7 materials-14-00904-f007:**
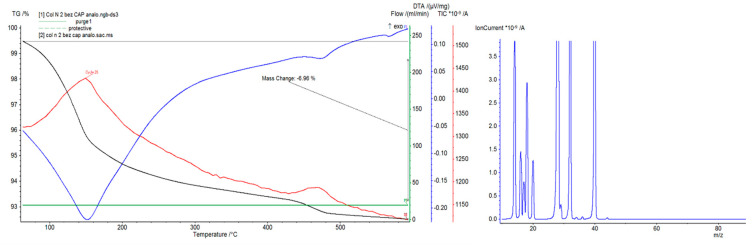
Thermal analysis DTA/DSC of sample 3.

**Figure 8 materials-14-00904-f008:**
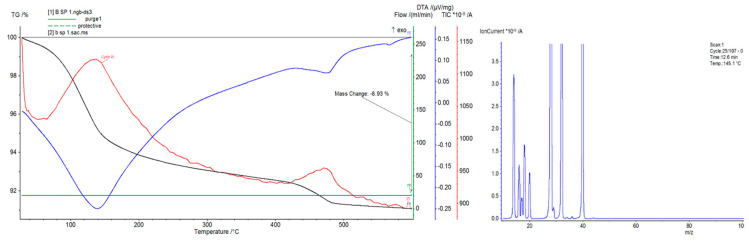
Thermal analysis DTA/DSC of sample 5.

**Figure 9 materials-14-00904-f009:**
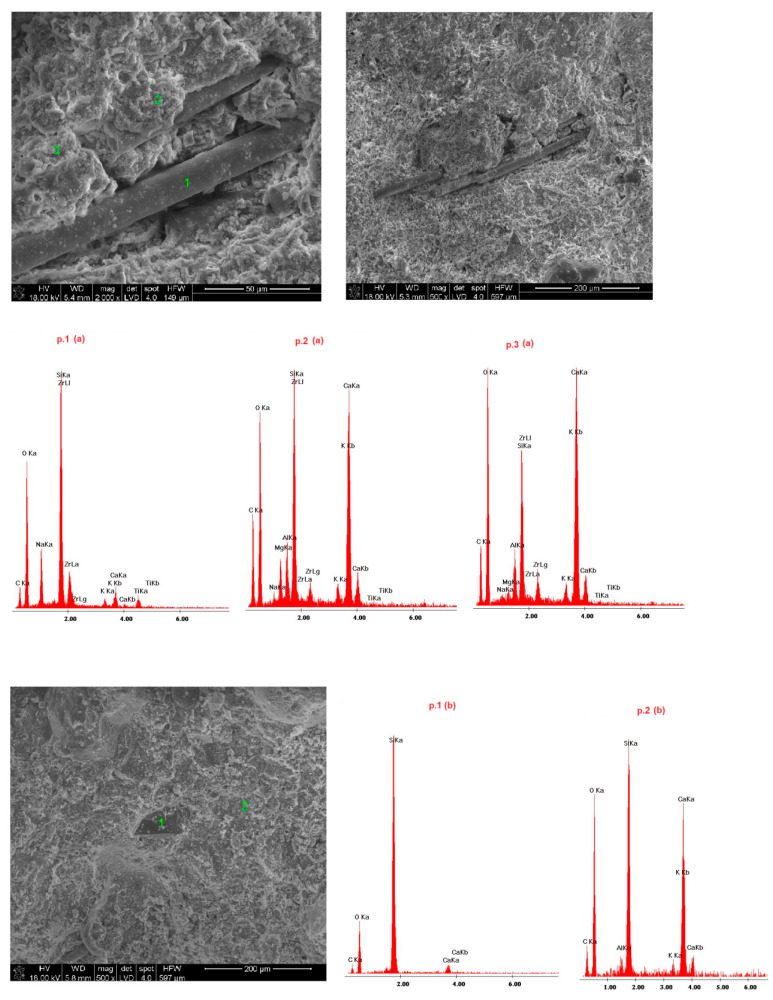
SEM microphotograph and energy dispersive spectroscopy (EDS) analysis of sample 1.

**Figure 10 materials-14-00904-f010:**
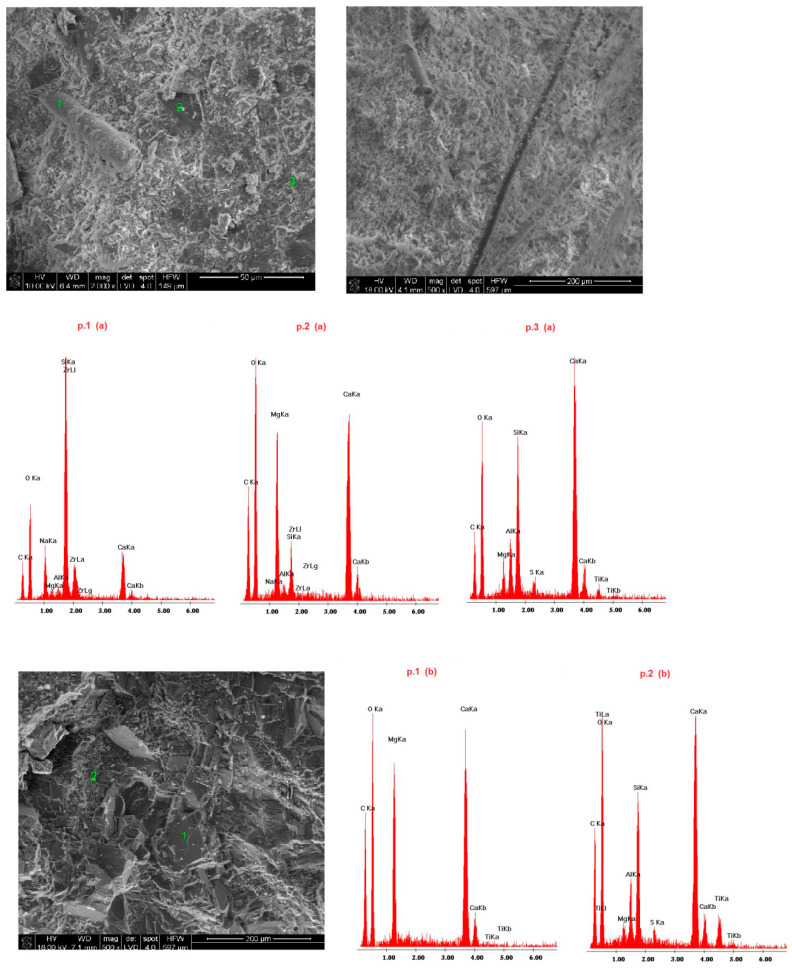
SEM microphotograph and EDS analysis of sample 2.

**Figure 11 materials-14-00904-f011:**
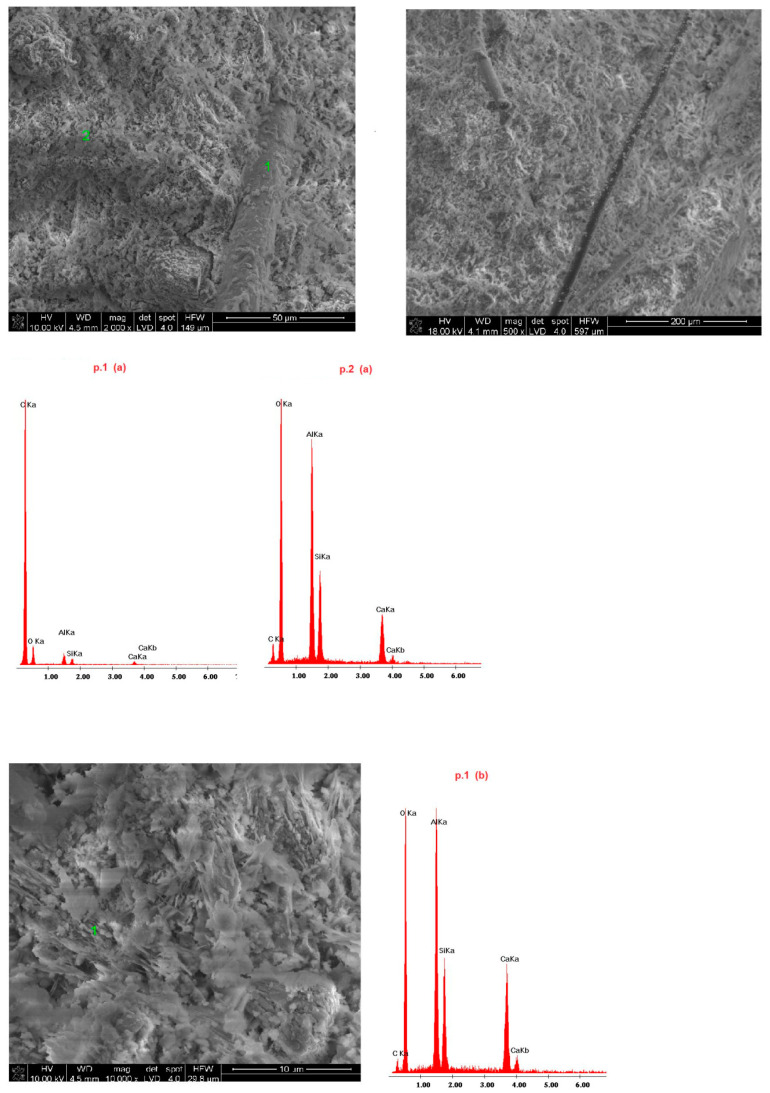
SEM microphotograph and EDS analysis of sample 3.

**Figure 12 materials-14-00904-f012:**
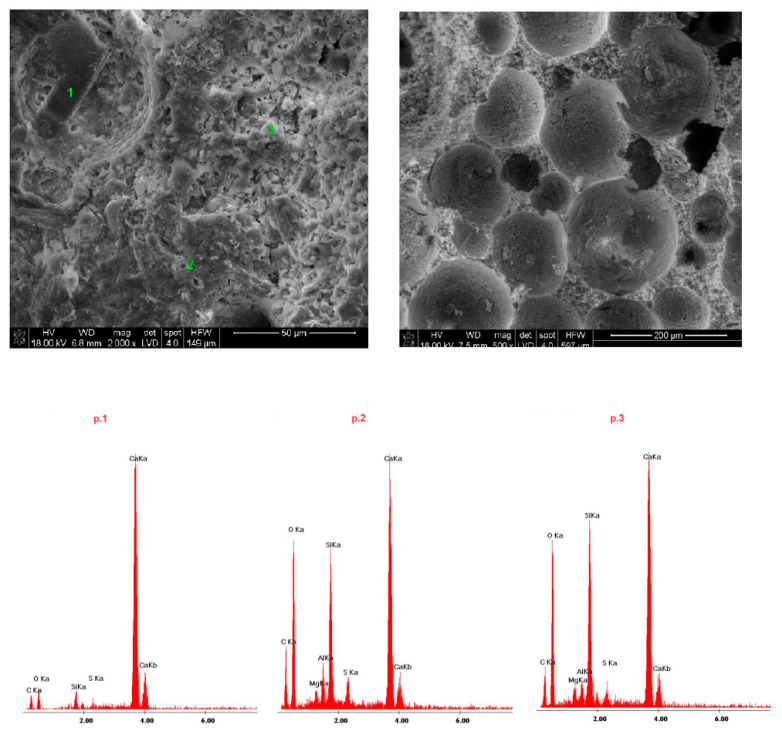
SEM microphotograph and EDS analysis of sample 4.

**Figure 13 materials-14-00904-f013:**
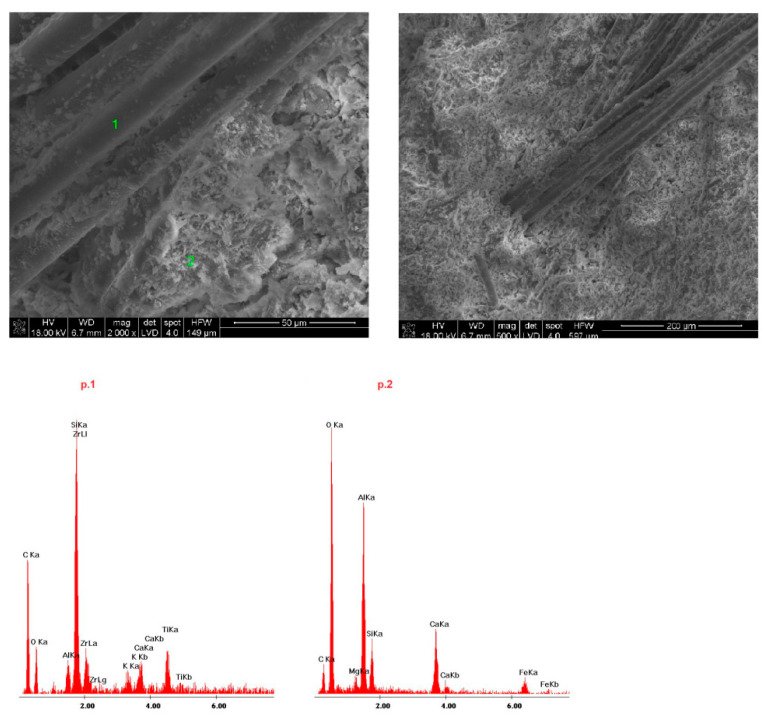
SEM microphotograph and EDS analysis of sample 5.

**Table 1 materials-14-00904-t001:** Concrete mix compositions used in research

Sample	Ingredient	% wt
1 Concrete with sandstone aggregate enriched with glass fibres	Portland cement CEM I 52.5 R Sand aggregate 0–2 mm Quatrz meal 0.75 Glass fibres 12 mm Pigment Superplasticizer + additives	28.528 45.645 13.313 0.266 0.799 1.654
2 Concrete with silicate and refractory aggregate enriched with polypropylene fibres	High alumina cement 70 Portland cement CEM I 52.5 R Fine silicate aggregate 0–2 mm Refractory aggregate 0–5 mm Polypropylene fibres 3 mm Pigment Superplasticizer + additives	15.1 4.9 12.0 67.4 0.141 0.304 0.100
3 Concrete with silicate and carbonate aggregates enriched with glass and polypropylene fibers	Portland cement CEM I 52.5 R Carbonate aggregate 8–16mm Silicate aggregate 0–4 mm Glass fibres 12 mm Polypropylene fibres 3 mm Pigment Superplasticizer + additives	31.343 47.062 6.268 0.209 0.020 1.567 2.037
4 Foam concrete with sand aggregate enriched with glass and steel fibers	Portland cement CEM I 52.5 R Sand 0–2 mm Glass fibres 12 mm Steel fibres 25 mm Superplasticizer + additives	26.725 61.824 1.158 0.677 1.242
5 Concrete with mafic agregate enriched with polypropylene and steel fibres	Portland cement CEM I 52.5 R High alumina cement 40 Mafic fine aggregate 0.75–3 mm Mafic aggregate 2–8 mm Polypropylene fibres 6 and 20 mm Steel fibres 12 mm Pigment Superplasticizer + additives	4.740 11.037 32.298 40.355 0.025 0.677 0.406 0.304

**Table 2 materials-14-00904-t002:** Physical parameters of the tested samples.

Sample	Mass (g)	Volume (cm^3^)	Density (g/cm^3^)
1	447.24	210.05	2.13
2	351.52	199.31	1.76
3	372.98	176.7	2.11
4	282	188.6	1.5
5	589.33	272.65	2.16

**Table 3 materials-14-00904-t003:** Temperature of the tested samples depending on the cooling time.

Time (min)	Temperature (°C)				
	1	2	3	4	5
0	134.8	131.8	128.6	132.1	137.7
1	130.7	125.4	119	127.3	124.6
2	126.4	121.6	115.6	121.7	122.5
3	123.5	116	111.7	113.8	118.3
4	119.5	110.6	109.5	110.4	115.8
5	115	106.2	103.6	104.5	111.2
6	112.6	102.3	101	100.1	109.5
7	108.1	98.5	96.6	96.7	104.3
8	104.4	94.6	90	88	103.3
9	100.9	91.2	88.3	86.1	99.9
10	97.8	86.9	87.8	80.2	95.2
12	91.2	81.6	86.2	73.6	91.6
14	85.9	74.6	79.3	68.5	87.6
16	79.5	71.5	71.5	64.2	82.6
18	76.5	67.5	68.8	56.7	77.2
20	72.2	63	65.6	53.4	76.9
25	63.4	53.4	58.9	46.9	74
30	57.2	47.6	54.3	42	67.4
35	52.8	43.3	50.3	38.4	58.4
40	48.8	38.7	48.6	38	54.6
45	45.5	37.2	46.1	36.2	52.6
50	40.9	36.4	45.8	35.8	48.1

**Table 4 materials-14-00904-t004:** Specific heat of raw materials used to prepare samples.

Ingredient	*c*_v_ (J/kg·K)
Sand	800
Cement	504
Water	4190
Silicate aggregate	1006
Mafic aggregate	990
Carbonate aggregate	920
Refractory aggregate	1750
Glass fiber	840
Polypropylene fiber	1460
Steel fiber	440

**Table 5 materials-14-00904-t005:** The total specific heat of the tested samples.

Sample	Σ *c*_v_ (J/kg·K)
1	1017.79
2	765.78
3	1170.03
4	990.58
5	1120.15

**Table 6 materials-14-00904-t006:** The ∆*T* and ∆*Q* values of the materials tested.

Sample	∆*T* (K)	∆*Q* (kJ)
1	93.9	42.72
2	95.4	25.64
3	82.8	36.13
4	96.3	26.9
5	89.6	59.11

**Table 7 materials-14-00904-t007:** Volume thermal capacity of the tested materials.

Sample	*b* (kJ/m^3^·K)	*b*_max_ (MJ/m^3^)
1	2167.89	203.57
2	1347.77	128.50
3	2468.76	204.43
4	1485.87	143.01
5	2419.52	216.83

**Table 8 materials-14-00904-t008:** Thermal power of the tested materials.

Sample	*P* (W)
1	14.24
2	8.55
3	12.04
4	8.97
5	19.70

**Table 9 materials-14-00904-t009:** Values of the mechanical strength under bending of the tested fireplace materials.

Sample Symbol	Maturation Time (days)		
	1	7	28
	Mechanical (MPa)		
1	8.5	9.9	12.3
2	3.5	4.3	5
3	7.4	10	13.2
4	2.9	4.1	4.7
5	1.8	3	4.8

## Data Availability

The data presented in this study are available on request from the corresponding author. The data are not publicly available due to the agreement with Northstar Poland.

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
