# Peer review of "Research on the Thermal Properties of Fireplace Concrete Materials Containing Various Mineral Aggregates Enriched by Organic and Inorganic Fibers"

_materials, 2021, doi:10.3390/ma14040904_

Round 1

Reviewer 1 Report

1) Please add the hyperlinks for figures and tables for easier review.

2) Numerical simulation has been widely applied to analyze the effect caused by aggregates on transport features of concrete, which should be included and discussed in the literature survey. The authors can refer to the following 5 recent publications:

  • LIU QF, FENG GL, XIA J, et al. Ionic transport features in concrete composites containing various shaped aggregates: a numerical study. Compos Struct, 2018, 183: 371–380.
  • MAO LX, HU Z, XIA J, et al. Multi-phase modelling of electrochemical rehabilitation for ASR and chloride affected concrete composites. Compos Struct, 2019, 207: 176–189.
  • ABYANEH SD, WONG HS, BUENFELD NR. Modelling the diffusivity of mortar and concrete using a three-dimensional mesostructure with several aggregate shapes. Comput Mater Sci, 2013, 78: 63–73.

3) In Line 60, it is better to remove the comma symbol. 

4) In Line 77, the symbol θ is lack of explanation. It seems something wrong with the numbers in the x-coordinate.

5) In Line 95, why bmax is the product of the volume heat capacity b and a certain period of time Δt.

6) In Line 123, the wording “heat of the medium” should be “the medium of heat”.

7)In Line 128, there is a spelling wrong in the title.

8) For Table 1, please pay attention to the layout. It would be better to place each ingredient in one line and making it correspond to its proportion. 

9) In Line 216, it would be better to add the ∑ in the right side of the equation.

10) In Lines 220 and Figures 6-8, there is something wrong with the spelling.

11) In line 254, why the thermal capacity and the density of metals is measured by the fraction?

12) Please explain DTA, TIC,TG etc. when the first mention.

13) In Line 276, more explanations are needed for the relationship between weight loss and the temperature increase.

14) In Line 316, micro-area a and micro-area b are not marked in the figure 9.

15) In Line 386 and Line 411, grammer mistakes. The word “and” should be replaced by but also.

16) In Line 389, wrong spelling for the wording “decohesive”.

17) In Lines 377-385 and the fourth paragraph of the discussion part, there are some repetitive explanations for the effects of fibers on bridging the cracks induced by stresses accompanying the hydration. It is better to combine these two paragraphs together.  

Author Response

Dear Sir/Madam

Thank you very much for the time and effort which let us to improve the quality of our article. The revision in the manuscript has been highlighted using green colour. Detailed explanations can be found in the attachment.

Yours sincerely,

Agata Stempkowska

On behalf of all the co-authors.

Reviewer 2 Report

The paper gives a thorough analysis of fireplace concrete made of fibres and mineral aggregates. The paper is beneficial to readers, but there are minor revisions needed to improve the quality of paper before publication:

Samples 1 -5: It would be better to give a notation to the samples based on their mix ratios.

Lines 220, 271, 280, 283: Translate in English

Table 9: Translate in English

Why were the mehanical properties determined? What is the signifcance in a fireplace?

Please re-write discussion section to include mechanical as well as thermal properties of differnt mixes.

In the conclusion, add which mix performed better for fire reistance.

Author Response

Dear Sir/Madam,

We are really grateful to your comments and I hope that our corrections to the manuscript will satisfy you. Detailed explanations can be found in the attachment.

On behalf on all co-authors

Yours faithfully,

Joanna Mastalska-Popławska

Round 2

Reviewer 1 Report

The authors have made a good effort to address the comments. The reviewer is happy to recommend it for publication.